# Interstitial Pneumonia with Autoimmune Features: Why Rheumatologist-Pulmonologist Collaboration Is Essential

**DOI:** 10.3390/biomedicines9010017

**Published:** 2020-12-26

**Authors:** Marco Sebastiani, Paola Faverio, Andreina Manfredi, Giulia Cassone, Caterina Vacchi, Anna Stainer, Maria Rosa Pozzi, Carlo Salvarani, Alberto Pesci, Fabrizio Luppi

**Affiliations:** 1Rheumatology Unit, University of Modena and Reggio Emilia, Azienda Ospedaliero-Universitaria Policlinico di Modena, 41124 Modena, Italy; marco.sebastiani@unimore.it (M.S.); Andreina.Manfredi@gmail.com (A.M.); cassonegiu@gmail.com (G.C.); caterina.vacchi@unimore.it (C.V.); carlo.salvarani@unimore.it (C.S.); 2Respiratory Unit, San Gerardo Hospital, Department of Medicine and Surgery, University of Milan-Bicocca, 20900 Monza, Italy; paola.faverio@unimib.it (P.F.); annetta.stainer@gmail.com (A.S.); alberto.pesci@unimib.it (A.P.); 3Rheumatology Unit, San Gerardo Hospital, 20900 Monza, Italy; m.pozzi@asst-monza.it; 4Rheumatology Unit, Dipartimento Medicina Interna e Specialità Mediche, Azienda Unità Sanitaria Locale di Reggio Emilia—Istituto di Ricerca e Cura a Carattere Scientifico, 42123 Reggio Emilia, Italy

**Keywords:** interstitial pneumonia with autoimmune features, connective tissue diseases, autoimmunity, interstitial lung diseases, idiopathic interstitial pneumonias, prognosis, classification, antibody

## Abstract

In 2015 the European Respiratory Society (ERS) and the American Thoracic Society (ATS) “Task Force on Undifferentiated Forms of Connective Tissue Disease-associated Interstitial Lung Disease” proposed classification criteria for a new research category defined as “Interstitial Pneumonia with Autoimmune Features” (IPAF), to uniformly define patients with interstitial lung disease (ILD) and features of autoimmunity, without a definite connective tissue disease. These classification criteria were based on a variable combination of features obtained from three domains: a clinical domain consisting of extra-thoracic features, a serologic domain with specific autoantibodies, and a morphologic domain with imaging patterns, histopathological findings, or multicompartment involvement. Features suggesting a systemic vasculitis were excluded. Since publication of ERS/ATS IPAF research criteria, various retrospective studies have been published focusing on prevalence; clinical, morphological, and serological features; and prognosis of these patients showing a broad heterogeneity in the results. Recently, two prospective, cohort studies were performed, confirming the existence of some peculiarities for this clinical entity and the possible progression of IPAF to a defined connective tissue disease (CTD) in about 15% of cases. Moreover, a non-specific interstitial pneumonia pattern, an anti-nuclear antibody positivity, and a Raynaud phenomenon were the most common findings. In comparison with idiopathic pulmonary fibrosis (IPF), IPAF patients showed a better performance in pulmonary function tests and less necessity of oxygen delivery. However, at this stage of our knowledge, we believe that further prospective studies, possibly derived from multicenter cohorts and through randomized control trials, to further validate the proposed classification criteria are needed.

## 1. Introduction

Interstitial lung diseases (ILDs) refer to a broad category of more than 200 lung diseases including a variety of illnesses with diverse causes, treatments, and prognoses. These disorders are grouped together because of similarities in their clinical presentation, plain chest radiographic appearance, and physiologic features leading ultimately—at least in a number of cases—to pulmonary fibrosis. ILDs contain several categories, characterized by different prognoses—including idiopathic interstitial pneumonias (IIPs) and connective tissue disease (CTD)-associated interstitial lung disease (CTD-ILD) [1].

The IIPs are a group of heterogeneous disorders characterized by diffuse parenchymal lung involvement with overlapping clinical and radiologic features [2]. They are generally categorized by histopathologic pattern, and the term chronic fibrosing interstitial pneumonia (IP) has recently been used to encompass the histopathologic patterns of usual interstitial pneumonia (UIP) and nonspecific interstitial pneumonia (NSIP) [1].

CTDs are a group of diseases with heterogeneous systemic features and possible immune-mediated, multi-organ dysfunction. The respiratory tract can be targeted, with different frequencies, in virtually every CTD and with a multitude of manifestations. However, among pulmonary manifestations, ILD is considered the most frequent and serious pulmonary complication, resulting in significant morbidity and mortality [3].

Distinguishing CTD-ILD from an IIP, specifically idiopathic pulmonary fibrosis (IPF), is of paramount importance because CTD-ILD has generally a more favorable prognosis and the available therapeutic options differ significantly [4].

However, in clinical practice, it is common to come across patients with an “idiopathic” interstitial pneumonia (IIP) associated with features suggestive of, but not diagnostic for, a classical CTD [5,6].

On the basis of previous studies, in 2015 the European Respiratory Society (ERS) and the American Thoracic Society (ATS) “Task Force on Undifferentiated Forms of Connective Tissue Disease-associated Interstitial Lung Disease” proposed classification criteria for a new research category defined as “Interstitial Pneumonia with Autoimmune Features” (IPAF). The classification of IPAF can therefore be considered an overlap between an idiopathic interstitial pneumonia and CTD-ILDs [7].

The aims of this review were to describe the evidence available regarding IPAF, including advantages and limitations of the current criteria, the implications for management, the future directions of this clinical entity, and the importance of a close collaboration between pulmonologists and rheumatologists.

## 2. Criteria for Interstitial Pneumonia with Autoimmune Features: The European Respiratory Society/American Thoracic Society Research Statement

As discussed above, there is agreement that some patients with an idiopathic ILD may have some features that suggest the presence of a systemic autoimmune process but do not meet classification criteria for a defined CTD [8,9,10]. Therefore, it is common to have discordance among specialists about how to diagnose such patients. A correct identification of patients with CTD-ILD can be challenging if the lung is the predominant or the primary organ involved and clinical evidence of a systemic autoimmune disease is subtle or absent [9].

Moreover, different IIPs may have different prognoses and the available therapeutic options may significantly differ [11,12,13].

In contrast to the adverse effects observed in patients with IPF [14], CTD-ILD could benefit from immunosuppressive treatment, including steroids [15,16]. On the other hand, patients with IPF benefit from antifibrotic agents [17,18], currently under investigation in CTD-ILD patients [19,20]. The lack of consensus over nomenclature and classification criteria limits the ability to perform prospective studies in patients with this particular subset of interstitial pneumonia.

Between 2010 and 2012, different studies proposed different, but partially overlapping, criteria and terms to describe these patients, including “undifferentiated CTD-associated ILD” (UCTD-ILD) [21,22], “lung-dominant CTD” [23] or “autoimmune-featured ILD” [24]. However, each term was capturing different patients and, therefore, none has been universally accepted.

The main limitation of these criteria was their poor applicability and heterogeneity. Studies using one set of criteria were not likely to be applicable to cohorts from different centers using other sets of criteria.

In an effort to systematically compare these four proposed criteria, Assayag and colleagues applied them to a cohort of 119 patients with ILD, showing that only 18% met all four definitions [25]. This study definitely showed that a uniform definition and nomenclature were needed to study these patients apparently affected by IIP but showing an “autoimmune flavor”. 

The term “interstitial pneumonia with autoimmune features” (IPAF) was proposed in 2015 by the ERS/ATS “Task Force on Undifferentiated Forms of Connective Tissue Disease-associated Interstitial Lung Disease” to describe individuals with interstitial pneumonia and features suggestive of an underlying systemic autoimmune condition who do not meet current criteria for a specific CTD [7].

The purpose of the statement was to standardize nomenclature and classification of such patients and to create a more uniform population in order to provide a platform for future research investigations and to elucidate prognostic and therapeutic dilemmas. The statement was not intended to provide criteria for the clinical management of these patients.

Several a priori requirements must be fulfilled for the classification of IPAF: Individuals must have evidence of interstitial pneumonia by high-resolution computed tomography (HRCT) imaging and/or by surgical lung biopsy; known causes for interstitial pneumonia must have been excluded after a thorough clinical evaluation; and patients must not meet criteria for a characterizable CTD. Then, the classification criteria combine features from three primary domains: (1) a clinical domain consisting of specific extra-thoracic signs or symptoms; (2) a serologic domain with specific circulating autoantibodies; (3) a morphologic domain with imaging and/or histopathological patterns and multicompartment involvement.

To be classified as having IPAF, patients must meet all of the a priori requirements and have at least one feature from at least two of the three domains (Table 1).

The included features were chosen for their higher specificity for CTDs. The term “connective tissue disease” was specifically avoided due to concerns that such labeling may give a false impression that these individuals have a defined CTD. In contrast, specific features suggesting vasculitis were deliberately excluded.

The Task Force included an international, multidisciplinary panel of CTD-ILD experts, including 13 pulmonologists, four rheumatologists, one thoracic radiologist, and one pulmonary pathologist. The proposed criteria reflected the panel’s expert opinion and Authors recognized that they should be tested and validated in future prospective studies, as a high-quality patient cohort for validation does not exist at the moment.

## 3. Retrospective Studies Applying ERS/ATS Research Criteria

Since publication of ERS/ATS IPAF research criteria, various retrospective studies have been published focusing on prevalence; clinical, morphological, and serological features; and prognosis of these patients showing conflicting results. (Table 2) [26].

Prevalence of IPAF among undifferentiated CTD or IIP patients range from 7.3% to 91% [27] and it is probably due to the heterogeneity of the populations screened.

A female predominance is frequently described [28], while smoking habit is variably reported [26,28,29,30,31,32].

Regarding the domains indicated by the ERS/ATS IPAF research statement, in the clinical domain, Raynaud phenomenon and arthritis or morning joint stiffness were the most represented features, while digital ulceration, telangiectasias, and Gottron sign are described at a smaller extent, probably due to the high specificity of these items for systemic sclerosis (SSc) and inflammatory idiopathic myopathies [31,32,33,34]. In contrast, sicca syndrome and serositis, which are not included in IPAF criteria, were described in a significant percentage of IPAF patients [31,32,33]. 

Anti-nuclear antibody (ANA) without any specific pattern or rheumatoid factor (RF) are the main immunological markers when serological domain is fulfilled, while the majority of other auto-antibodies, highly specific for definite CTDs, are very rare or totally absent.

In the morphological domain, radiologic patterns considerably varied among case series, particularly regarding the UIP pattern, which is differently represented according to the study considered, with some showing a relatively high prevalence of the UIP pattern, such as in the Oldham’s study, in which it is estimated in about 45% of the cohort [29]. This is a major difference with CTD-ILD, whose pathological or radiological findings are historically dominated by NSIP and NSIP-organizing pneumonia (OP) overlap [35]. 

However, the UIP pattern was not included as a specific morphologic feature in the IPAF statement since, in a patient with interstitial pneumonia, the presence of UIP pattern alone does not increase the likelihood of having CTD, with the primary aim to reduce the inclusion of patients with IPF. In this regard, some Authors suggested to exclude patients with UIP pattern from IPAF classification [35]. However, this is in contrast with the high prevalence of UIP pattern in some rheumatic diseases, such as rheumatoid arthritis or Sjogren’s syndrome, and the ERS/ATS IPAF research statement does not exclude categorization as IPAF in patients with UIP [7].

Nevertheless, retrospective studies applying ERS/ATS research criteria showed a broad heterogeneity within the IPAF phenotype, with substantial prognostic implications. 

These studies also pointed to a need for revisions to the IPAF criteria, particularly with respect to the morphologic domain.

Data regarding prognosis are also rather heterogeneous [26]. A systematic review by Kamiya and Panlaqui, exploring prognostic factors in IPAF, highlighted the inconsistency of IPAF prognosis in comparison to IPF or CTD-ILD. Although prognosis seems to be related to the prevalence of UIP pattern, with similar results between patients with IPAF or IPF with a UIP pattern [36], the multivariate analysis showed that only age is a prognostic factor for all-cause mortality in IPAF patients [36]. 

In regard to functional ventilatory progression, Kim et al. observed a slower decline of diffusion capacity for carbon monoxide (DLCO) and total lung capacity (TLC) in IPAF group compared to non-IPAF-IIP group, but no differences were observed in terms of forced vital capacity (FVC) decline rate [37].

Acute exacerbation of ILD (AE-ILD) is considered a life-threatening complication of IPF [38]. In a retrospective study by Lim and colleagues, AEs in IPAF were significantly less frequent than in IPF and CTD-ILD, being respectively observed in 25.9%, 32.9%, and 35.4% of subjects with a statistically significant difference between groups [32]. Consistently, Yoshimura and colleagues reported that IPAF was an independent protective factor for AE-ILD [30]. 

Moreover, in some studies, an association between the fulfilling of clinical domain [29,30] and a better overall survival has been described, while multicompartment features within the morphological domain were a strong predictor of poor outcome [29]. 

Finally, IPAF may represent an early phase of a CTD; in this case, the evolution into definite CTD is more likely during the first years of follow-up [10]. 

A combination of female gender and the fulfilling of serological domain predicted the progression in 50 patients initially classified as IPAF [39].

## 4. Prospective Studies

The main aim of the ERS/ATS IPAF statement is to provide a platform for defining research criteria for prospective studies. In fact, this statement is not intended as a guideline for clinical care. On the basis of this issue, two prospective, cohort studies have been already published.

In the study published by Sambataro and colleagues [34], 626 consecutive ILD patients were evaluated over a two-year time: 45 IPAF patients were compared with a cohort of 143 IPF patients, showing that IPAF had a predominance of female gender (62.12%) with a median age of 66 years. The most common findings included a prevalence of NSIP (68.89%) in the morphological domain, the presence of anti-nuclear antibody positivity (17.77%) regarding the serological domain, and Raynaud phenomenon (31.11%) in the clinical domain. In comparison with IPF, IPAF patients showed younger age, better performances in pulmonary function tests, less necessity of oxygen support, and predominance of female gender and NSIP pattern.

In our study [33], 52 IPAF patients were consecutively enrolled and prospectively followed up. The clinical domain for IPAF was satisfied in 44 patients, showing a prevalence of joint stiffness and Raynaud phenomenon. Distal digital ulceration, palmar telangiectasias, digital edema, and Gottron sign were not observed in any patient. Moreover, despite not being included in the IPAF criteria, sicca syndrome and serositis were recorded in a significant percentage of patients. Serological domain was satisfied in 94.2% of patients and, in particular, anti-nuclear antibodies (ANA) were detected in 72.3% of patients. No patients showed anti-DNA or anti-citrullinated peptides antibodies (ACPA). Finally, the morphological domain for IPAF classification was recorded in 55.8%. In particular, UIP pattern at HRCT was described in 44.2%, NSIP in 32.7%, and OP in 15.4%, while 7.7% was not classifiable. Of the patients, 13.5% developed a definite CTD during the follow-up. The estimated five-year survival for IPAF patients was compared with 104 IPF patients, showing a significantly higher survival in IPAF compared to IPF.

These studies confirmed that IPAF appears to be considered as a separate clinical entity with clinical, radiological, pathological, and survival differences compared to IPF. Similarities between the two studies occurred regarding age, gender, and characteristics of the three domains.

An interesting difference between these two studies is the inclusion of patients with a UIP pattern. In the study by Sambataro et al., no IPAF patients with a UIP pattern were included. In contrast, our study enrolled IPAF patients with a UIP pattern. Unfortunately, the different study design does not allow any comparison between them regarding survival since the presence of a UIP pattern is considered a poor prognostic factor in many studies.

## 5. Advantages and Limitations of the Current ATS/ERS Criteria

Since the publication of the ERS/ATS IPAF research statement, various Authors have debated about their advantages and limitations and many proposals of revision have been made [42,43,44,45,46].

The original sin of IPAF criteria is their use in clinical practice, out of their aim of “research criteria”. In fact, no validation studies have been performed subsequently to their publication, but they have been broadly used in clinical practice and a clinical diagnosis of IPAF is routinely made [43,44].

The main advantage of IPAF criteria is their facility of use, easily applicable from both rheumatologist and pulmonologist. In particular, pulmonologists might be encouraged to investigate the presence of a rheumatic disorder and to increase their awareness about the heterogeneity of autoimmune diseases as possible cause of ILD; in contrast, rheumatologists may challenge themselves with the various ILD patterns diagnosed in these patients.

Furthermore, for the first time, a shared definition has been suggested to fill a gap in the identification of these patients, previously corresponding to a grey zone in diagnosis and management. Pulmonologists included IPAF patients among IIPs, often overlooking their possible extrapulmonary complications. On the other side, rheumatologists usually classified these patients as undifferentiated CTD, even when no other organ involvement was recognizable [47].

Finally, the research statement of the ERS/ATS IPAF statement underscored the opportunity of a multidisciplinary evaluation for diagnosis of interstitial pneumonias [7], highlighting a major change in the diagnostic approach to diffuse lung disease. This concept is largely accepted by pulmonologists [2], but, for the first time, also rheumatologists are actively involved in multidisciplinary discussion (MDD) [48,49].

However, many criticisms have been addressed to IPAF criteria. First, a large heterogeneity of IPAF patients have been reported by many Authors [26,28,29,31,40,43].

Moreover, it is common to notice that some patients classified as IPAF could be considered to have incomplete or early forms of CTD or overlap with the traditional definitions of UCTD [5,26,33].

Finally, it is likely that some individuals who initially were considered to have IPAF will progress over time to a defined CTD. Therefore, these patients might be initially categorized as “IPAF” inappropriately [26,33].

CTDs are a very large family of diseases and, together with rheumatoid arthritis, have significant peculiarities both in treatment and in extrapulmonary and lung manifestations.

The IPAF classification might support the choice of an immunosuppressive treatment in some patients [10,50,51]. This therapeutic approach could be reviewed after the results of the INBUILD study, which has shown the efficacy of nintedanib also in CTD and IPAF patients with progressive fibrosing ILD [19].

On the other side, the future availability of antifibrotic drugs also for CTD and rheumatoid arthritis (RA) patients could reduce the level of attention in differential diagnosis between CTDs and IIPs, also in consideration of the poor prognosis (similar to IPF) of patients categorized as affected by IPAF with a radiological and/or pathological UIP pattern. This point is crucial. In fact, although in patients with isolated lung involvement, both idiopathic or autoimmune-mediated, the therapeutic approach could be the same [19], a correct diagnosis is essential to guarantee an appropriate follow-up to patients who may developed a specific CTD. In fact, CTDs can remain asymptomatic for a long period of time with possible unpredictable flares; in this context, the lack of a definite diagnosis might cause a delay in treatment and a worse prognosis.

Although lung involvement is frequently observed in rheumatic diseases, it has not been included in the classification criteria (often incorrectly used for diagnosis) of CTDs, with the exclusion of systemic sclerosis (SSc) [52]. Therefore, rheumatologists are not confident in suggesting a diagnosis of CTD in patients with isolated lung involvement [47,53].

Moreover, clinical and classification criteria not always are coincident, and IPAF criteria may include patients usually differently diagnosed by pulmonologist and rheumatologist [35]. In this regard, anti-synthetase syndrome (ASSD) and SSc are two archetypal examples. SSc can develop without skin involvement; when lung involvement is predominant and classification criteria are not satisfied, a diagnosis of IPAF might be made [54]. ASSD is a heterogeneous condition characterized by the clinical triad ILD, myositis, and arthritis, but incomplete disease is frequent, with positive anti-tRNA-synthetase antibodies and only one or two major clinical manifestations. In this setting, a patient with a positive anti-tRNA-synthetase antibody and ILD could be considered to have ASSD by some clinicians and IPAF by others, despite the high risk of developing other clinical findings of the classic triad [55,56]. In consideration of this possible overlap, currently the CLASS (Classification Criteria of Anti-synthetase Syndrome) project is ongoing to develop specific classification criteria for ASSD [57]

SSc and ASSD are two paradigmatic examples, but the same concept may be extended for almost all CTDs and RA with a predominant or primary lung involvement [58,59].

On the other side, antineutrophil cytoplasmic antibodies (ANCA) were excluded from the IPAF classification as they are associated with systemic vasculitis and not with CTD. However, the combination of ILD and ANCAs, in particular anti-myeloperoxidase (MPO) specificity, shares with IPAF many issues related to the overlap between ILD and a chronic systemic disease [33].

In conclusion, the current definition of IPAF does not appear to be able to select homogenous populations and it probably excludes other categories of patients. Therefore, IPAF can be considered a bridge between IIP and CTD-ILDs, useful in the research setting but not yet to be considered as a specific clinical disease entity.

In contrast, in the context of follow-up, IPAF classification is useful in selecting subsets of ILD patients at risk of developing a defined CTD. Multidisciplinary evaluation remains essential, including rheumatologist and pulmonologist, also during follow-up to early identify evolution in a CTD.

## 6. The Importance of the Multidisciplinary Discussion

Earlier guidelines regarding ILDs recommended collaboration between specialists of different domains being clinicians, radiologists, and histopathologists to work together in order to establish a confident diagnosis of IIPs [2].

The approach of MDD has also been tested by Flaherty and colleagues [60], showing that—in IPF diagnosis—the MDD involving pulmonologists, radiologists, and pathologists improves the inter-observer agreement in comparison with the initial diagnosis made by the individual expert. In fact, HRCT of the chest shows a high inter-observer variability [61,62]. Also, histopathology—considered in the past as the “gold standard” in ILDs’ diagnosis—is characterized by a high inter-observer variability and potential sampling error [63,64].

Clinical, radiological, and pathological data taken individually can be a source of bias. For example, a patient with a UIP pattern on imaging can be diagnosed as having IPF, whereas the patient can present with extrapulmonary clinical features suggestive of RA and should consequently be diagnosed as RA-ILD [49].

Recently, a prospective study evaluated the contribution of rheumatological assessment to ILD differential diagnosis, through a MDD, by comparing the diagnosis before and after the rheumatological evaluation. As predicted, the addition of routine rheumatological evaluation significantly altered the final diagnosis, emphasizing the importance of rheumatologists as expert figures [65].

In conclusion, we believe that a multidisciplinary approach may be important in the diagnostic workup of IPAF with the aim to improve diagnostic confidence, specifically versus IIPs and CTD-ILDs. In contrast to earlier studies, this MDD should include an expert rheumatologist with the aim to better interpret serologic autoimmunity and clinical signs and symptoms, particularly when they are subtle.

## 7. Implications for Management

The current treatment for patients classified as IPAF is not well established. Non-IPF ILD treatment is, per se, challenging and it is generally based on the use of steroids and immunosuppressants, while antifibrotic drugs, namely pirfenidone and nintedanib, are the mainstream of IPF treatment. Non-pharmacologic treatment includes pulmonary rehabilitation, oxygen supplementation, education, and palliative care.

No randomized controlled trials or case-control studies have been specifically designed for evaluating efficacy and safety of different immunosuppressive agents in IPAF patients. Only retrospective studies are available, suggesting the use of mycophenolate mofetil (MMF) and azathioprine as first-line agents in these patients, mainly according to previous experience with CTD-associated ILDs [50,66,67,68,69,70,71], while rituximab and calcineurin inhibitors should be proposed in “non-responder” cases.

McCoy et al. retrospectively reviewed 52 patients who met criteria for IPAF, 28 receiving MMF and 24 not treated with immunosuppressants (median time treatment, 22 months): No differences were found in pulmonary function tests (PFT) change or mortality in patients exposed to MMF compared to unexposed patients. Furthermore, findings of the study suggested that MMF could be more effective in patients with more ground-glass opacities and less reticulation on HRCT [51].

Five patients with IPAF were included in another large series of patients with different refractory ILDs treated with rituximab, following prednisone with or without MMF treatment. Among them, four patients showed stability or improvement of their pulmonary function after treatment with rituximab, while one patient died [72].

New possible perspectives have been suggested by more recent evidence about the use of antifibrotic drugs in patients with progressive fibrosing ILD. As previously cited, the INBUILD study aimed to investigate the efficacy and safety of nintedanib in patients with fibrosing interstitial lung diseases with a progressive phenotype; it also enrolled unclassifiable interstitial pneumonia, including IPAF patients. Results showed a significant lower annual rate of FVC decline in patients who received nintedanib rather than placebo, as the primary outcome [19].

A multicenter, double-blind, randomized, placebo-controlled phase 2 trial investigated the efficacy and safety of oral pirfenidone in progressive fibrosing unclassifiable ILD. The patients were stratified according to the concomitant mycophenolate mofetil use and presence or absence of interstitial pneumonia with autoimmune features. Compared with the placebo group, patients in the pirfenidone group showed a lower median change in FVC and were less likely to have a decline in FVC of more than 5% or 10%. No differences in total and serious adverse events were recorded between pirfenidone and placebo groups [73].

Clearly, large prospective trials are needed in these patients to ascertain whether there is a benefit to using immunosuppression, and what agent could be the most effective.

## 8. Conclusions

The term IPAF describes patients with ILD who have an “autoimmune flavor” without a characterizable CTD. Before IPAF, diverse criteria and definitions had been proposed to characterize such patients, capturing different patients’ phenotypes.

The recent ERS/ATS research statement provided a uniform nomenclature and criteria, triggering various retrospective studies with the aim to better characterize this undefined group of ILD patients. Particularly, they showed the broad heterogeneity between IPAF patients and the impact on prognosis of the underlying lung pattern. Longitudinal follow-up is required because some patients evolve to a defined CTD.

Moreover, two prospective, cohort studies have been performed, confirming the existence of some peculiarities for this clinical entity and the possible progression to a defined CTD in about 15% of cases. Moreover, a NSIP pattern, an ANA positivity, and a Raynaud phenomenon were the most common findings. In comparison with IPF, IPAF patients showed a better performance in pulmonary function tests and less necessity of oxygen supplementation.

However, at this stage of our knowledge, we believe that further prospective studies, possibly derived from multicenter cohorts and through randomized controlled trials to further validate the proposed classification criteria are needed.

## Figures and Tables

**Table 1 biomedicines-09-00017-t001:** Classification criteria for interstitial pneumonia with autoimmune features.

A priori requirements:		
Presence of an interstitial pneumonia by HRCT or SLB and		
Exclusion of alternative aetiologies and		
Does not meet criteria for a defined CTD and		
At least one feature from at least two of the following domains:		
**A. Clinical domain**	**B. Serologic domain**	**C. Morphologic domain**
Distal digital fissuring (i.e., “mechanic hands”)	ANA ≥ 1:320 titer, diffuse, speckled, homogeneous patterns or	Suggestive radiology patterns by HRCT:
Distal digital tip ulceration	a. ANA nucleolar pattern (any titer) or	NSIP
Inflammatory arthritis or polyarticular morning joint	b. ANA centromere pattern (any titer)	OP
stiffness ≥60 min	Rheumatoid factor ≥2x upper limit of normal	NSIP-OP overlap
Palmar telangiectasia	Anti-CCP	LIP
Raynaud phenomenon	Anti-dsDNA	Histopathology patterns or features by surgical lung biopsy:
Unexplained digital oedema	Anti-Ro (SS-A)	NSIP
Unexplained fixed rash on the digital extensor surfaces	Anti-La (SS-B)	OP
(Gottron sign)	Anti-ribonucleoprotein	NSIP-OP overlap
	Anti-Smith	LIP
	Anti-topoisomerase (Scl-70)	Interstitial lymphoid aggregates with germinal centres
	Anti-tRNA synthetase (e.g., Jo-1, PL-7, PL-12; others are: EJ, OJ, KS, Zo, tRS)	Diffuse lymphoplasmacytic infiltration
		Multicompartment involvement (in addition to interstitial pneumonia):
	Anti-PM-Scl	
	Anti-MDA-5	Unexplained pleural effusion or thickening
		Unexplained pericardial effusion or thickening
		Unexplained intrinsic airways disease (includes airflow obstruction, bronchiolitis or bronchiectasis)
		Unexplained pulmonary vasculopathy

**Table 2 biomedicines-09-00017-t002:** Comparison of retrospectively and prospectively identified interstitial pneumonia with autoimmune features (IPAF) cohorts.

	Oldham et al. [29], 2016	Chartrand et al. [28], 2016	Ahmad et al. [31], 2017	Ito et al. [40], 2017	Dai et al. [41], 2018	Yoshimura et al. [30], 2018	Kelly & Moua [35], 2018	Lim et al. [32], 2019	Alevizos et al. [39], 2019	Kim et al. [37], 2020	Sambataro et al. [34], 2019	Sebastiani et al. [33], 2020
Patients	144	56	57	98	177	32	101	54	50	109	45	52
Study design	Retrospective	retrospective	retrospective	retrospective	retrospective	Retrospective	retrospective	Retrospective	retrospective	retrospective	prospective	prospective
Age, y (mean ± SD)	63.2 ± 11	54.6 ± 10.3	64.4 ± 14	67.5 ± 9	67.6 ± 8.6	63.4 ± 12.6	56.9 ± 14.2	67.9 ±10.5	56 (47–44) ^d^	60.6 ± 11.6	66 (59.5–71) ^d^	68 (54–82) ^d^
Female	52.1	71.4	49.1	58.2	55.9	40.6	39	64.8	60	56	62.2	55.8
Ever smoker	54.9	32.1	34	38.8	19.2	56.2	31	27.8	50	63.3	51.1	63.5
Clinical	49.3	62.5	47.3	NR	20.3	53.1	NR	31.5	38	25.7	62.2	84.6
Serologic	91.7	91.1	93	100 ^a^	92.1	71.9	NR	90.7	98	100	48.9	94.2
Morphologic	85.4	98.2	78.9	100 ^b^	95.5	96.9	NR	81.5	92	72.5	100	55.8
Clinical and serologic	14.6	2	NR	NR	NR	3.1	4	NR	NR	25.7	NR	NR
Clinical and morphologic	8.3	9	NR	NR	NR	28.1	14	NR	NR	17.4	51.1	NR
Serologic and morphologic	50.7	37.5	NR	100	NR	46.9	26	NR	NR	72.5	37.8	NR
All 3 domains	26.4	52	NR	NR	NR	21.9	56	NR	NR	17.4	11.1	
UIP by HRCT	54.6	8.9	28	0	4.5	NR	11.9	25.9	18	36.7	0	44.2
Underwent SLB, n (%)	83 (57.6)	36 (64.3)	16 (28.1)	17 (17.3)	0 ^c^	22 (68.8)	51 (50.5)	NR	40 (80)	NR	NR	2 (3.8)
UIP on SLB, n (%)	61 (73.5)	8 (22.2)	3 (18.8)	3 (17.6)	-	-	12 (23.5)	NR	20 (40)	NR	NR	2 (3.8)
Treatment												
Corticosteroids	32.2	81.8	67.9	17.3	72.3	59.4	NR	NR	NR	84.4	NR	63.5
Antifibrotic	NR	NR	5.4	2	NR	25	NR	NR	0	0	NR	11.5
Outcome												
Death	39.6	0	12.3	27.6	19.8	NR	28	27.8	NR	NR	2.2	28.8
Lung transplant	10.8	NR	NR	NR	NR	NR	NR	NR	NR	NR	NR	NR

Data presented as % unless otherwise stated; NR: not reported; y: years; SD: standard deviation; UIP: usual interstitial pneumonia; HRCT: high-resolution computed tomography; SLB: surgical lung biopsy; ^a^: based on study design, inclusion criteria was positive serological evaluation; ^b^: based on reported HRCT findings of nonspecific interstitial pneumonia (NSIP), organizing pneumonia (OP), or NSIP + OP in 98 of 98 subjects; ^c^: all histopathology from transbronchial biopsies; ^d^: median.

## Data Availability

Not applicable.

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
