# Peer review of "Interstitial Pneumonia with Autoimmune Features: Why Rheumatologist-Pulmonologist Collaboration Is Essential"

_biomedicines, 2020, doi:10.3390/biomedicines9010017_

Round 1
Reviewer 1 Report
This is a well-written review of an important topic in the field of ILD (IPAF).
The review is well constructed and informative.
I have just a comment regarding UIP Pattern. UIP Pattern was not included in the morphological criteria of IPAF definition because in a patient with interstitial pneumonia, the presence of a UIP pattern alone does not increase the likelihood of having CTD. Nonetheless, several retrospective series cited in this review do describe patients with a UIP pattern. UIP pattern is seen in many CTD-ILD (As RA or Sjogren s.)
I would like to read some comments from the Authors about this apparent contradiction
Author Response
R1.C1
This is a well-written review of an important topic in the field of ILD (IPAF).
The review is well constructed and informative.
R1.R1
We thank the Reviewer for appreciating our work.
R1.C2
I have just a comment regarding UIP Pattern. UIP Pattern was not included in the morphological criteria of IPAF definition because in a patient with interstitial pneumonia, the presence of a UIP pattern alone does not increase the likelihood of having CTD. Nonetheless, several retrospective series cited in this review do describe patients with a UIP pattern. UIP pattern is seen in many CTD-ILD (As RA or Sjogren s.)
I would like to read some comments from the Authors about this apparent contradiction
R1.R2
This is an important point and we thank the Reviewer for raising it. In the section entitled “Retrospective studies applying ERS/ATS research criteria”, we comment about the apparent discrepancy between the IPAF statement and the following retrospective studies regarding the presence of a UIP pattern in the framework of a patient satisfying IPAF criteria.
Reviewer 2 Report
This is a comprehensive review of the classification criteria for interstitial pneumonia with autoimmune features (IPAF) proposed by the European Respiratory and the American Thoracic Societies in 2015. These criteria are generally used in clinical practice although they were created with a research purpose. Authors also make a review of demographic and clinical characteristics of patients diagnosed with IPAF reported in different retrospective and prospective studies. They conclude that nonspecific interstitial pneumonia, anti-nuclear antibody positivity and Raynaud’s phenomenon were the most common findings. However, other manifestations like palmar telangiectasias or Gottron’s sign, included in IPAF classification criteria, were unusual. Moreover, sicca syndrome, serositis or usual interstitial pneumonia radiological pattern, not included in IPAF criteria, were found in a considerable number of cases. Multidisciplinary evaluation including pulmonologists and rheumatologists is essential to optimize management of, in this case, patients with IPAF. This matter is noted and justified throughout the manuscript.
Comments and suggestions for the Authors:
The manuscript is well written and compiles extensive and updated information on IPAF and the requirement of further validation of the proposed classification criteria, with an imperative collaboration between rheumatologists and pulmonologist.
I do not have any major concern point.
I would only suggest mentioning the following two papers that could fully complete the evidence of IPAF characteristics:
- Atienza-Mateo B, Remuzgo-Martínez S, Mora Cuesta VM, Iturbe-Fernández D, Fernández-Rozas S, Prieto-Peña D, Calderón-Goercke M, Corrales A, Blanco Rodríguez GB, Gómez-Román JJ, González-Gay MÁ, Cifrián JM. The Spectrum of Interstitial Lung Disease Associated with Autoimmune Diseases: Data of a 3.6-Year Prospective Study from a Referral Center of Interstitial Lung Disease and Lung Transplantation. J Clin Med. 2020 May 26;9(6):1606.
- Cavagna L, Gonzalez Gay MA, Allanore Y, Matucci-Cerinic M. Interstitial pneumonia with autoimmune features: a new classification still on the move. Eur Respir Rev. 2018 Jun 27;27(148):180047.
Author Response
R2.C1
The manuscript is well written and compiles extensive and updated information on IPAF and the requirement of further validation of the proposed classification criteria, with an imperative collaboration between rheumatologists and pulmonologist.
I do not have any major concern point.
R2.R1
We thank the Reviewer for appreciating our work.
R2.C2
I would only suggest mentioning the following two papers that could fully complete the evidence of IPAF characteristics:
- Atienza-Mateo B, Remuzgo-Martínez S, Mora Cuesta VM, Iturbe-Fernández D, Fernández-Rozas S, Prieto-Peña D, Calderón-Goercke M, Corrales A, Blanco Rodríguez GB, Gómez-Román JJ, González-Gay MÁ, Cifrián JM. The Spectrum of Interstitial Lung Disease Associated with Autoimmune Diseases: Data of a 3.6-Year Prospective Study from a Referral Center of Interstitial Lung Disease and Lung Transplantation. J Clin Med. 2020 May 26;9(6):1606.
- Cavagna L, Gonzalez Gay MA, Allanore Y, Matucci-Cerinic M. Interstitial pneumonia with autoimmune features: a new classification still on the move. Eur Respir Rev. 2018 Jun 27;27(148):180047.
R2.R2
We thank the Referee for his/her suggestion. We have added these two very interesting references in the text.